# Nutritional Status and Quality of Life in Hospitalised Cancer Patients Who Develop Intestinal Failure and Require Parenteral Nutrition: An Observational Study

**DOI:** 10.3390/nu12082357

**Published:** 2020-08-07

**Authors:** Marina Plyta, Pinal S. Patel, Konstantinos C. Fragkos, Tomoko Kumagai, Shameer Mehta, Farooq Rahman, Simona Di Caro

**Affiliations:** 1Intestinal Failure Service, Department of Gastroenterology, University College London Hospitals NHS Foundation Trust, London NW1 2BU, UK; marina.plyta@gmail.com (M.P.); pinal.patel1@nhs.net (P.S.P.); konstantinos.fragkos@nhs.net (K.C.F.); shameer.mehta@nhs.net (S.M.); farooq.rahman@nhs.net (F.R.); 2Division of Medicine, University College London, London WC1E 6BT, UK; tomoko.kumagai.17@ucl.ac.uk

**Keywords:** nutrition, quality of life, advanced cancer, oncology, parenteral nutrition

## Abstract

(1) Background: Malnutrition in cancer patients impacts quality of life (QoL) and performance status (PS). When oral/enteral nutrition is not possible and patients develop intestinal failure, parenteral nutrition (PN) is indicated. Our aim was to assess nutritional status, QoL, and PS in hospitalised cancer patients recently initiated on PN for intestinal failure. (2) Methods: The design was a cross-sectional observational study. The following information was captured: demographic, anthropometric, biochemical and medical information, as well as nutritional screening tool (NST), patient-generated subjective global assessment (PG-SGA), functional assessment of cancer therapy-general (FACT-G), and Karnofsky PS (KPS) data. (3) Results: Among 85 PN referrals, 30 oncology patients (56.2 years, 56.7% male) were identified. Mean weight (60.3 ± 16.6 kg) corresponded to normal body mass index values (21.0 ± 5.1 kg/m^2^). However, weight loss was significant in patients with gastrointestinal tumours (*p* < 0.01). A high malnutrition risk was present in 53.3–56.7% of patients, depending on the screening tool. Patients had impaired QoL (FACT-G: 26.6 ± 9.8) but PS indicated above average capability with independent daily activities (KPS: 60 ± 10). (4) Conclusions: Future research should assess the impact of impaired NS and QoL on clinical outcomes such as survival, with a view to encompassing nutritional and QoL assessment in the management pathway of this patient group.

## 1. Introduction

Cancer is among the leading causes of morbidity and mortality worldwide [1]. In advanced disease, malnutrition and involuntary weight loss with associated cachexia are observed in up to 80% of patients [2,3]. Half of all cancer deaths worldwide are attributed to malignancies, with a high prevalence of cachexia, such as gastrointestinal and pulmonary malignancies. However, cachexia is also highly prevalent at the end of life regardless of tumour type [4]. Systemic inflammation caused by the underlying malignant process drives cachexia’s progression through metabolic disturbances and muscle loss [4,5]. Other factors such as anorexia and cancer-related symptoms that may reduce oral intake, side-effects of medications and therapies, as well as functional impairment and psychological distress also contribute to cachexia [4,5,6]. The prevalence of muscle loss is reported in 20–70% of cancer patients, depending on tumour site and stage, and is predictive of prognostication, highlighting the importance of halting cachexia progression [5].

Disease progression and its treatment often severely impact patients’ nutritional status and quality of life (QoL) [5]. Historically, the identification of patients at risk of malnutrition relied on weight loss and body mass index (BMI) trajectory since diagnosis [7]. However, patients are often subjected to fluid imbalance [8] and/or may suffer from obesity prior to their cancer diagnosis [4]. Thus, confounding factors limit the utility of traditional anthropometric measures as a reliable marker of nutritional status (NS) [9]. Other anthropometric measurements that take into consideration skeletal muscle loss and strength, such as mid arm circumference (MAC), mid upper arm muscle circumference (MUAMC), hand grip strength (HGS) and triceps skinfold thickness (TSF), are used alongside weight to concurrently assess body composition [1]. The drive to include more reliable body composition assessment methods to quantify lean mass has led to an increase in the use of computerised tomography (CT) and magnetic resonance imaging (MRI) scans as tools of nutritional assessment [10]. Bioelectrical impedance (BIA) and ultrasound are non-invasive methods, also utilised as prognostic tools in research, but use in clinical practice warrants consideration of fluid distribution disturbances [8]. The use of nutritional screening tools to stratify patients into malnutrition risk categories and triaging the dietetic referral processes also remain part of the standard practice, yet are subject to confounding factors, as most of them incorporate weight assessment as part of their constructs [4].

The European Society for Clinical Nutrition and Metabolism (ESPEN, Luxembourg) suggests early initiation of nutritional support after diagnosis [1]. Artificial nutrition (AN) might be required to ensure that the nutritional requirements are met in cases of inadequate food intake. However, in advanced stages, it is not always feasible to deliver AN orally or enterally [7,11]. Thus, in such cases where intestinal failure develops, parenteral nutrition (PN) may be indicated [12]. As PN is not without additional risks and costs to the patients (e.g., complex medical management, intravenous catheter care, and risk of serious infections), its provision needs to be carefully considered only in those with estimated survival of more than two months, willing to undergo this treatment option, and importantly willing to cope with the practicalities that go with it [13,14,15,16].

There is a surplus of research indicating that patients receiving PN in inpatient settings or at home (home parenteral nutrition (HPN)) have improved NS, QoL, reduced chemotherapy toxicity and prolonged survival [9]. However, the effect of PN in patients with advanced cancer remains unclear [13,14]. Impaired NS is independently and adversely associated with QoL, function and ultimately prognostication in cancer patients, and inflammation appears to be the underpinning cause [17]. Malnourished patients are more likely to experience psychological distress due to prolonged hospital stay, readmissions, loss of control and independence, fatigue and sense of helplessness, as well as poor performance status (PS) caused by weight loss and altered physical appearance, ultimately impairing overall QoL [4,7]. Validated questionnaires assess QoL in this patient population [18,19]. However, the majority of studies have been performed in home settings [20], not taking into account the adverse effects of prolonged hospitalisation or acute health conditions. Thus, although it has been suggested that oncology patients with short-term inpatient PN provision might report improved QoL [20], this improvement may be in part due to the benefits of the multimodal therapy that they receive during hospitalisation [21].

Hence, the aim of this study is to assess NS, PS and QoL in hospitalised cancer patients where PN was recently initiated due to intestinal failure. We believe that the present study will provide a comparative view of nutritional screening tools and assessments in this patient cohort seeking to improve clinical practice.

## 2. Materials and Methods

### 2.1. Study Setting

This study was conducted at University College London Hospital (UCLH, London, UK) a 665-bed tertiary hospital and a national referral centre for a wide range of cancers, in central London, UK. PN provision services are supported by the multidisciplinary nutrition support team, which reviews all inpatient PN referrals identified by the primary clinical teams.

### 2.2. Study Population

All adult patients, aged 18 years or older, admitted to UCLH with active cancer, referred to the nutrition support team and started on PN between January and June 2019 were screened for inclusion in this study. Patients who were adults with active cancer, had capacity to complete questionnaires, consented for additional anthropometric measures, and were not established on HPN prior to their present admission were included. The design was a cross-sectional observational study. Data were collected through UCLH patient records (paper and electronic) close to the time of referral and during patient interviews (Appendix A).

### 2.3. Data Collection

#### 2.3.1. Demographics and Serum Biochemistry

Demographics and medical information: gender, age, primary malignancy, metastases, surgery, chemotherapy and/or radiotherapy before or during PN, and indication for PN (ESPEN) [12].

Serum biochemistry: C-reactive protein (CRP), white cell count (WCC), albumin (Alb), haemoglobin, sodium (Na), potassium (K), magnesium (Mg), phosphate (PO_4_), adjusted calcium (AdjCa), urea, and creatinine.

#### 2.3.2. Anthropometrics, NS, QoLand PS

Anthropometrics and NS: height, weight upon starting PN, habitual body weight, habitual and current BMI, percentage of weight loss upon starting PN from habitual weight, MAC [22], TSF [22], HGS [23], and MUAMC. The latter was calculated with the following equation MUAMC = MAC − (0.314 ∗ TSF) [24]. MUAMC is strongly correlated with whole-body composition assessments (e.g., BIA and dual-energy x-ray absorptiometry) in similar populations [25]. Patients were defined as having cancer cachexia if weight loss > 5% was reported in the past 6 months since diagnosis or BMI ≤ 20 Kg/m^2^ and any degree of weight loss > 2% [2].

The UCLH nutritional screening tool (NST) is performed by trained staff (health care assistant, nurse or dietitian) to screen patients within 24 h of admission and weekly thereafter in order to identify those in need of dietary support (Appendix A). The NST assesses patients’ weight, height, BMI, appetite, dietary intake, weight loss, psychological and neurological status, and physical appearance, while guiding triaging for the appropriate actions required. The NST categorises patients into low, medium and high malnutrition risk groups, with higher scores indicating a greater malnutrition risk (scores 0–2, 3–6, >7, respectively). Medium- and high-risk patients require a dietetic referral.

The patient-generated subjective global assessment-short form (PG-SGA) is a validated nutrition assessment tool for cancer patients designed for self-administration, focusing on weight and food intake changes, symptoms that have persisted for more than 2 weeks, as well as changes in activities and performance. The PG-SGA categorises patients as well nourished, moderately or suspected of being malnourished or severely malnourished, with higher scores indicative of greater malnutrition risk (scores 0–1, 2–8, >9, respectively) [26].

QoL: The functional assessment of cancer therapy-general (FACT-G) is a self-administered 27-item questionnaire. Trained staff (dietitians) calculated the overall (0–108) and subscale scores using specific guidelines (higher scores indicated better QoL) [19]. Permission to use the FACT-G questionnaire was obtained from website [27].

The HPN-QoL questionnaire is specifically designed for oncology patients treated with HPN [28]. This questionnaire was adapted, excluding the HPN-specific questions, and used as a measure of QoL for patients who had started inpatient PN. Permission to use the HPN-QoL questionnaire was sought from the author (Janet Baxter).

PS: The Karnofsky PS (KPS) was assessed by the attending dietitian. Scores range from 0 to 100, with over 50 indicating that the patient is unable to carry out daily tasks, but able to live at home and care for most personal needs with varying amounts of assistance [29,30]. The Eastern Cooperative Oncology Group/World Health Organisation performance status (ECOG/WHO-PS) is a prognostic factor in cancer populations, using a scale from 0 (fully active) to 5 (dead) to indicate the ability of physical activity, movement and self-care [31].

### 2.4. Ethical Considerations

This was a cross-sectional study and the principles of the Declaration of Helsinki were followed during design and analysis. Ethical approval was not required for this study as it was registered as a departmental audit.

### 2.5. Statistical Analysis

Data are presented as the mean (standard deviation [SD]) or frequencies and percentages. Univariate analyses were conducted with the chi-square test, Spearman’s rho for correlations, *t*-tests and ANOVA. The concordance between tools of malnutrition risk assessment, namely the NST, the PG-SGA, BMI and weight loss %, was examined using Cohen’s kappa for categorical variables [32]. Exploratory factor analysis based on principal component analysis was used for the HPN-QoL questionnaire for item aggregation and reduction [33], and internal consistency was examined using Cronbach’s alpha coefficient [34]. A cluster heatmap with a dendrogram was also produced based on hierarchical clustering. Statistical significance was reported at *p* < 0.05. All statistical analyses were performed using IBM SPSS Statistics (Release 25.0.0.1 2017, Chicago (IL), USA: SPSS, Inc., an IBM Company) and R 4.0 (R Foundation for Statistical Computing, Vienna, Austria).

## 3. Results

### 3.1. Descriptives

Descriptives regarding patient characteristics, serum biochemistry, anthropometrics, NS, QoL and PS are shown in Table 1 and Table 2. Among 85 patients referred to the nutrition support team, 30 patients satisfied inclusion criteria (females 43.3%, 13/30; mean age 56.2 (16.4) years). Most patients had upper gastrointestinal and haematological tumours (both 36.7%, 11/30), with advanced disease present in approximately half of these patients (46.7%, 14/30) (Table 1). Cancer cachexia was present in 70% (18/30) of patients. Patients presented with low mean QoL, as indicated by mean FACT-G score (26.6 ± 9.8), but a PS which indicated above average capability with independent daily activities (KPS: 60 ± 10) (Table 2).

### 3.2. Nutritional Status

#### 3.2.1. Prevalence of Malnutrition

According to the NST, 20.0% (5/25) were at low risk of malnutrition and 80.0% (20/25) were at risk, of which 16.0% (4/25) were at medium and 64% (16/25) were at high risk of malnutrition. According to the PG-SGA, 100% were malnourished, of which 29.2% (7/24) were moderately or suspected of being malnourished and 70.8% (17/24) were severely malnourished. When malnutrition was classified according to weight loss % and BMI, only 42.3% (11/26) had weight loss more than 10% and 34.5% (10/29) had a BMI less than 20 kg/m^2^, indicating risk of malnutrition (Table 1).

#### 3.2.2. Agreement between Malnutrition Risk Assessment Tools

Compatibility between the different nutritional screening tools was assessed with Cohen’s kappa and there was no significant compatibility between any tool (*p* > 0.05) (Appendix A). The NST was moderately correlated with MUAMC (rho = −0.426) and weight on admission (rho = −0.455), and moderately with weight loss at 6 months (rho = 0.502) (Appendix A).

#### 3.2.3. Nutritional Status Indicators

Nutritional indexes were examined according to the type of malignancy, indication for PN, presence of metastases and cachexia and are presented in Table 3. There was a trend towards a significant difference in weight upon starting PN according to the primary cancer location, with upper gastrointestinal cancer patients having a lower weight than other types of malignancy (48.1 ± 10.0 kg, *p* < 0.01). This patient group also had lower baseline BMI values (17.22 ± 4.5 kg/m^2^, *p* < 0.01). Additionally, patients with cachexia had significantly higher NST scores (cachexia 3.9 (3.5) vs. no cachexia 8.2 (4), *p* < 0.05).

Nutritional indexes were next examined according to NST, PG-SGA, KPS and WHO-PS scores and are presented in Table 3. There were no significant results observed, except for patients’ HGS according to KPS score. Patients in the low KPS scores category (score < 50) had lower HGS measurements (24.6 ± 9.22 kg), while those in the high KPS scores category (score ≥ 50) had higher HGS measurements (44.1 ± 22.6 kg, *p* < 0.05).

The association between nutritional indexes was examined using Spearman’s rho correlation (Appendix A). All correlations were significant (*p* < 0.05). TSF was moderately correlated with BMI on admission (rho = 0.531), weight on admission (rho = 0.469), and weight loss at 6 months (rho = 0.425). MAC was moderately correlated with BMI on admission (rho = 0.553) and strongly with weight on admission (rho = 0.683). MUAMC was very strongly correlated with BMI on admission (rho = 0.805) and weight on admission (rho = 0.830) and moderately with weight loss at 6 months (rho = 0.475). Finally, HGS was moderately correlated with weight on admission (rho = 0.386), MAC (rho = 0.440), and KPS (rho = 0.531).

Presence of cachexia and metastasis, NST and PG-SGA score distributions were examined by PN indication, type of malignancy, metastatic disease and KPS score, and are shown in Appendix A. Only presence of metastatic disease differed by types of primary malignancy, with patients affected by upper gastrointestinal malignancies having a higher prevalence of metastatic disease (*p* < 0.05).

### 3.3. Quality of Life

#### 3.3.1. Exploratory Factor Analysis and Internal Consistency

Factor analysis of the results of the HPN-QoL questionnaire identified five factors, based on the Kaiser criterion [35], also confirmed by the line drop after the fifth component in the Cattell scree plot (Appendix A) [36]. Appendix A shows the allocation of the items to the respective factors based on their highest factor loading, above the predefined value of 0.4 [37]. Cronbach’s alpha for the modified HPN-QoL questionnaire overall was 0.63. When internal consistency was performed separately for the factors, Cronbach’s alpha was over 0.80 in all factors. The FACT-G subscales report in a range of poor to good internal consistency, with Cronbach’s alpha ranging from 0.567 to 0.840 (Table 4).

#### 3.3.2. Quality of Life Correlations

The QoL questionnaire FACT-G, its subscales, and the factors of the modified HPN-QoL questionnaire were examined according to type of malignancy, presence of metastases and cachexia, and no statistically significant difference was observed (Appendix A). The aforementioned were also examined according to classification based on NST, PG-SGA, KPS and WHO-PS scores and no significant differences were detected (Appendix A). Only the scores of the energy/independence factor of the modified HPN-QoL questionnaire within the risk of malnutrition groups based on PG-SGA score had a non-statistically significant trend to differ, with severely malnourished patients having lower scores of energy/independence (0.6 ± 1.1 vs. −0.3 ± 0.8, *p* = 0.053).

The association between the QoL questionnaires and their subscales with NS indexes was examined using Spearman’s rho correlation and the significant results appear on Appendix A. While neither the FACT-G nor its subscales correlated with any of the nutritional indexes, two of the factors of the HPN-QoL questionnaire did. The pain factor correlated moderately with the HGS on the non-dominant hand (rho = −0.474), and the energy/independence factor correlated moderately with age (rho = 0.525).

### 3.4. Performance Status

Performance status was examined according to the type of malignancy, indication for PN, presence of metastases and cachexia, but there was no significant difference in patients’ KPS scores (Table 3). However, when the KPS was examined according to classification based on NST, PG-SGA and WHO-PS scores (Table 3), there was a significant difference in KPS scores between the subgroups within WHO-PS, suggesting that patients in the distinct WHO-PS subgroups had different KPS scores (73.3 ± 5.7 at WHO-PS 1, 64.6 ± 7.8 at WHO-PS 2, 55.4 ± 8.7 at WHO-PS 3, 40.0 at WHO-PS 4, *p* < 0.001). Finally, the KPS was moderately correlated with length of stay (rho = −0.448) (Appendix A).

### 3.5. Cluster Heatmap

A cluster heatmap for the NS, PS and QoL indexes was undertaken and the degree of similarity is presented in a dendrogram, where four clusters were identified (Figure 1).

## 4. Discussion

This is the first study in the literature examining nutritional status, QoL and PS in an inpatient setting for advanced cancer patients referred for PN due to intestinal failure. The PG-SGA identified 100.0% (24/24) of patients as being at risk of malnutrition and 70.8% (17/24) as having severe malnutrition, the majority of which were gastrointestinal and haematological cancer patients on active oncological treatment. Prevalence from studies with similar groups of patients ranged between 45.1% and 80.4% [38,39,40,41]. Results differed when the NST was used, as 80.0% (20/25) of patients were at risk, for whom close monitoring was required.

The PG-SGA was next compared to other malnutrition screening tools, with differences being noted. Compared to the NST, prevalence was similar for severely malnourished patients, though differences were observed between moderately and well-nourished patients. The PG-SGA is the reference tool for oncology patients’ nutritional assessment and is a sensitive, comprehensible, easy and quick tool to assess malnutrition [42,43]. Correcting for short-term improvements in weight and scoring multiple nutrition impact symptoms, it offers high accuracy in distinguishing well-nourished from malnourished patients; however, it ultimately categorises more patients at risk compared to other tools [44,45]. To the best of our knowledge, there have not been other studies assessing nutritional risk in cancer patients using the NST; therefore, comparison of the prevalence of patients identified as at risk of malnutrition using the NST with similar studies is disadvantaged.

Examination of malnutrition risk according to weight loss % and BMI classification failed to identify all patients at risk in our sample, with weight loss % and BMI identifying only 42.3% (11/26) and 34.5% (10/29) of patients, respectively. Previous studies evaluating weight loss and BMI as markers of NS in cancer patients corroborate this discrepancy, attributing such an observation to sudden changes in weight (e.g., ascites, oedema, and fluid retention) or inability to detect differences between fat and fat-free mass [8,46,47].

Patients with upper gastrointestinal malignancies were identified to have lower weight and BMI on admission compared to patients with other types of malignancies. This finding is consistent with other studies and is attributed to the nature and location of oesophageal and gastric cancers, where the risk of malnutrition is highly prevalent by the time of diagnosis, mainly due to bowel obstruction and gastrointestinal symptoms [48,49,50,51]. A borderline difference in MUAMC was noted between different types of malignancy, indicating that severe weight loss due to muscle mass depletion defines this pattern of malnutrition [52,53]. There was a trend for patients with metastatic disease to have a lower BMI compared to patients with no metastatic disease, indicative of cancer cachexia [54].

MAC, HGS, and TSF were significantly correlated with weight and BMI on admission. HGS was moderately correlated with weight on admission and MAC. Muscle mass and strength loss was also noted in our malnourished patients. Previous studies have not found significant results between HGS and weight loss, although lower HGS values have been noted in malnourished patients [55,56,57,58]. Low HGS reflects muscle mass depletion, which is reportedly due to altered protein metabolism during severe weight loss, inflammation, inactivity, anaemia, fatigue and tissue hypoperfusion [55,59]. Next, there was a significant difference in patients’ HGS between subgroups based on KPS scores. Patients with a poorer PS had significantly lower HGS measurements compared to patients with a higher PS, a trend noted in other studies as well [55,60,61,62]. PS is an important indicator of a patient’s ability to be functional and perform activities of daily living. The finding that decreased muscle strength could predict functional decline in hospitalised oncology patients is of utmost importance and also carries a prognostic value [14]. PS also reflects several behavioural patterns that are affected by NS.

The energy/independence factor from the HPN-QoL questionnaire was moderately correlated with age. This suggested that older patients felt less independent and more unable to cope with daily life and their illness, accounting for their poor functional status and QoL. Similar results have been reported in other studies [63,64], and a recent meta-analysis reported that 37–55% of older cancer patients required daily assistance [65]. The pain factor from the HPN-QoL questionnaire was moderately correlated with patients’ HGS, suggesting that lower HGS was related to a greater sense of pain. Pain is one of the most frequently reported symptoms by cancer patients, as well as one of the most important drivers of diminished appetite [66]. Pain, along with systemic inflammation, contributes to the sense of fatigue, which restricts physical activity and in turn alters protein metabolism and favours muscle wasting [7]. Finally, the dendrogram revealed that QoL clustered closely with PS and HGS, indicating that reduced muscle strength and consequently impaired PS, were the main components that compromised a patient’s QoL. Our study replicates this result in line with other studies which have also noted that QoL deteriorates alongside PS reduction in advanced cancer patients [66,67,68].

The main strength of the present study is the thorough assessment of NS, PS and QoL, beyond that of standard clinical assessment, with all measurements performed using validated instruments by experienced health care professionals with adequate training. In terms of limitations, the small study sample and relatively heterogenous cohort of patients with regards to type of malignancy led to reduced power in identifying significant findings. Secondly, the assessment was performed at a single point in time and follow-up information was not available, thus it is not possible to draw adequate conclusions on the causality of relationships. Finally, the lack of whole-body composition measures (e.g., CT/MRI scans) limits the amount of high-quality and reliable data that could be used as the gold standard to compare methods of nutritional assessment.

The present study offers certain implications for practitioners worth discussing. Firstly, since the NST identified patients with malnutrition and proportionately categorised them as having cachexia, further studies or audits could validate the NST in oncology patients, correctly identifying those in need of comprehensive dietetic assessment and detect periodic NS changes after nutritional support with PN. Furthermore, our results strengthen the notion that weight loss and BMI should be assessed assiduously as a sensitive, convenient and non-invasive assessment of cachexia. The next important point is that although PN was consistently used in these patients, muscle mass loss is unlikely to be reversed in refractory cachexia [2]. If loss of muscle mass cannot be prevented during hospitalisation and PS is ultimately compromised, discharge timelines are affected. Finally, since pain and loss of muscle mass have shown to be strongly correlated, multimodal care involving resistance training is strongly advocated in this group of patients [69]. An implication for research would be to investigate the impact of PN on improving the sense of pain, and consequently maintaining patients’ muscle strength, as part of the multimodal care.

## 5. Conclusions

Malnutrition in cancer patients is still under-recognised and highly prevalent. PN is an intricate therapy that should be judiciously used when indicated. However, it only comprises one part of the multimodal therapeutic strategies of cancer patients including psychological support, symptom management, anticancer therapy and physical activity. Our results highlight the compromised overall status of patients by the time of referral for PN support, hence timely referral as well as concurrent assessment of NS, PS and QoL in this patient group is of paramount importance due to interplays identified among them across the literature. Our aim is to raise awareness on the importance of preventing cachexia and PS and QoL deterioration among hospital staff and change our clinical practice towards appropriate individualised holistic patient-centred care plans, from which patients will derive maximal benefit.

## Figures and Tables

**Figure 1 nutrients-12-02357-f001:**
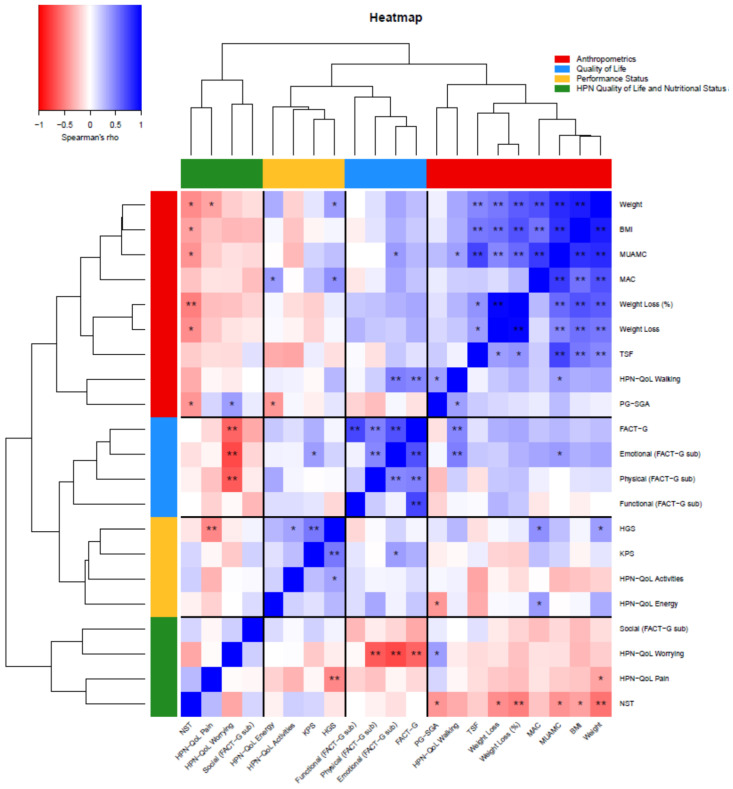
Cluster heatmap to represent Spearman’s rho correlations between variables. Each cell is coloured based on the level of relationship. Red indicates negative correlations while blue indicates positive correlations, and darker colours indicate stronger relationships while brighter colours indicate weaker relationships or the absence of relationships. * = level of statistical significance at 0.05, ** = level of statistical significance at 0.01. The four clusters were: anthropometrics (red), quality of life (blue), performance status (yellow), HPN-QoL and nutritional status (green). BMI = body mass index, FACT-G = functional assessment of cancer therapy-general, HGS = hand grip strength, KPS = Karnofsky performance status, MAC = mid arm circumference, MUAMC = mid upper arm muscle circumference, NST = nutritional screening tool, PG-SGA = patient-generated subjective global assessment, HPN-QoL = home parenteral nutrition quality of life questionnaire, TSF = tricep skinfold thickness, and WHO-PS = World Health Organisation performance status.

**Table 1 nutrients-12-02357-t001:** Patients’ characteristics (categorical variables).

	N (%)
Age (years)	
≤49	8 (26.7)
50–64	12 (40.0)
≥65	10 (33.3)
Gender	
Female	13 (43.3)
Male	17 (56.7)
Admission present complaint	
Chemotherapy related	3 (10.0)
Disease progression	17 (56.7)
Elective admission	10 (33.3)
Type of malignancy *	
Gynaecological	2 (6.7)
Upper gastrointestinal	11 (36.7)
Lower gastrointestinal	4 (13.3)
Haematological	11 (36.7)
Other	2 (6.7)
Metastatic disease	
No	16 (53.3)
Yes	14 (46.7)
Location of metastases	
No Metastases	16 (53.3)
Lower diaphragm	8 (26.7)
Upper diaphragm	4 (13.3)
Both	2 (6.2)
Surgery for malignancy	
No	10 (55.6)
Yes	8 (44.4)
Chemotherapy before/during PN	
No	3 (13.6)
Yes	19 (86.44)
Radiotherapy before/during PN	
No	8 (66.7)
Yes	4 (33.3)
Indication for PN	
Extensive small bowel mucosal disease	12 (40.0%)
Intestinal dysmotility	4 (13.3)
Mechanical obstruction	13 (43.3)
Short bowel syndrome or intestinal fistula	1 (3.3)
% weight loss upon starting PN	
≤5%	8 (30.8)
5–10%	7 (26.9)
10–15%	4 (15.4)
≥15%	7 (26.9)
BMI upon starting PN (kg/m^2^)	
BMI ≤ 20.0	10 (34.5)
BMI > 20.0	19 (65.5)
Cancer cachexia	
No	9 (30.0)
Yes	21 (70.0)
NST score	
Low risk	5 (20.0)
Medium risk	4 (16.0)
High risk	16 (64.0)
PG-SGA score	
Moderately malnourished	7 (29.2)
Severely malnourished	17 (70.8)
KPS score	
<50	9 (30.0)
50–100	21 (70.0)
WHO PS	
1	3 (10.0)
2	13 (43.3)
3	13 (43.3)
4	1 (3.3)
Referral to palliative care	
No	11 (57.9)
Yes	8 (42.1)
Line for PN	
PICC	30 (100.0)

BMI = body mass index, KPS = Karnofsky performance status, NST = nutritional screening tool, PG-SGA = patient-generated subjective global assessment, PN = parenteral nutrition, and WHO-PS = World Health Organisation performance status. * Gynaecological: breast and endometrial; lower gastrointestinal: small bowel, colon, and sigmoid; upper gastrointestinal: oesophageal and gastric; haematological: leukaemia, lymphoma, and amyloidosis; other: penis and bladder.

**Table 2 nutrients-12-02357-t002:** Patients’ characteristics (continuous variables).

	N	Mean (SD)
Anthropometrics		
Age (years)	30	56.2 (16.4)
Usual weight (kg)	26	71.4 (15.8)
Height (m)	29	1.7 (0.1)
Usual BMI (kg/m^2^)	25	24.1 (4.6)
Weight upon starting PN (kg)	30	60.3 (16.6)
Weight loss upon starting PN (%)	26	–20.0 (21.9)
BMI upon starting PN (kg/m^2^)	29	21.0 (5.1)
HGS on non-dominant Hand (kg)	27	38.2 (21.4)
Loss of HGS on non-dominant hand (%)	27	−2.7 (43.7)
MAC (cm)	27	20.3 (4.1)
TSF (mm)	27	12.3 (9.0)
MUAMC (cm)	27	24.2 (5.1)
Serum Biochemistry		
CRP (mg/L)	14	44.2 (79.2)
WCC (×10^9^/L)	19	9.4 (5.0)
Albumin (g/L)	19	36.8 (8.0)
Haemoglobin (g/L)	19	113.4 (23.7)
Sodium (mmol/L)	19	136.5 (5.0)
Potassium (mmol/L)	19	4.0 (0.7)
Magnesium (mmol/L)	13	0.8 (0.1)
Phosphate (mmol/L)	17	0.9 (0.3)
Adjusted calcium (mmol/L)	17	2.4 (0.1)
Urea	19	5.3 (1.9)
Creatinine	19	62.9 (17.0)
Nutritional Status		
NST	25	7.0 (4.1)
PS-SGA	25	12.4 (5.1)
Performance Status		
KPS	30	60.7 (10.5)
Quality of Life		
FACT-G	24	26.6 (9.8)
Physical well-being	24	11.0 (5.8)
Social/family well-being	24	22.8 (5.6)
Emotional well-being	24	14.7 (6.2)
Functional well-being	24	10.8 (6.6)
Length of stay	29	44.0 (24.3)

BMI = body mass index, CRP = C-reactive protein, FACT-G = functional assessment of cancer therapy-general, HGS = hand grip strength, KPS = Karnofsky performance status, MAC = mid arm circumference, MUAMC = mid upper arm muscle circumference, NST = Nutritional screening tool, PG-SGA = patient-generated subjective global assessment, PN = parenteral nutrition, TSF = tricep skinfold thickness, WCC = white cell count, and WHO-PS = World Health Organisation performance status.

**Table 3 nutrients-12-02357-t003:** Nutritional status and quality of life according various variables.

	Age (years)	Weight upon Starting PN (kg)	Weight Loss upon Starting PN (%)	BMI upon Starting PN (kg/m^2^)	HGS	MAC	TSF	MUAMC	NST Score	PG-SGA	KPS	Length of Stay
**Type of Malignancy**												
Gynaecological	61.9 (12.6)	61.4 (7.8)	−11.6 (3.7)	23.8 (4.3)	20.7 (7.8)	21.1 (1.1)	21.6 (9.3)	27.8 (1.8)	8.0 (2.8)	23.0 (.)	65.0 (7.1)	41.0 (18.4)
Upper gastrointestinal	63.3 (12.1)	48.1 (10.0)	−11.9 (10.3)	17.22 (4.5)	32.6 (14.6)	18.4 (4.4)	6.8 (5.9)	20.5 (5.7)	8.75 (4.6)	11.1 (3.9)	58.2 (9.8)	51.0 (28.8)
Lower gastrointestinal	55.8 (25.2)	60.7 (12.3)	−9.7 (5.0)	20.8 (3.1)	49.2 (23.3)	20.9 (3.4)	13.3 (10.2)	25.0 (2.3)	8.33 (1.5)	12.7 (3.2)	70.0 (8.2)	35.7 (9.7)
Haematological	48.1 (16.1)	66.5 (15.6)	−10.4 (13.1)	23.4 (3.1)	40.9 (26.4)	21.5 (4.1)	15.9 (9.3)	26.5 (3.9)	5.6 (4.0)	12.7 (6.3)	57.3 (11.0)	37.6 (19.0)
Other	56.4 (18.9)	86.2 (22.6)	−1.8 (2.8)	28.0 (7.6)	61.8 (.)	24.3 (.)	9.5 (.)	27.3 (.)	4.0 (5.6)	11.0 (7.0)	70.0 (0.0)	57.0 (52.3)
*p*-value	0.301	0.004	0.815	0.005	0.365	0.401	0.093	0.054	0.418	0.334	0.135	0.644
**Indication for PN**												
Extensive small bowel mucosal disease	56.7 (20.2)	58.8 (17.6)	−25.0 (27.8)	20.5 (4.9)	39.4 (26.9)	20.8 (4.8)	13.1 (10.6)	25.0 (5.2)	5.3 (4.4)	11.9 (4.5)	56.7 (11.5)	42.6 (21.9)
Intestinal dysmotility	59.5 (12.9)	64.9 (11.7)	−18.7 (22.0)	24.0 (3.3)	38.3 (16.0)	20.8 (3.8)	11.7 (8.2)	24.5 (6.3)	9.0 (1.4)	10.7 (2.7)	60.0 (8.2)	46.5 (32.4)
Mechanical obstruction	54.9 (15.2)	60.0 (18.5)	−16.6 (18.1)	20.7 (5.8)	34.0 (15.8)	19.8 (3.0)	12.6 (8.6)	23.4 (5.2)	8.0 (4.1)	13.9 (6.4)	63.8 (9.6)	47.1 (28.1)
Small bowel syndrome or intestinal fistula	54. 4 (.)	64.0 (.)	14.1 (.)	20.7 (.)	78.8 (.)	24.4 (.)	4.1 (.)	25.7 (.)	8.0 (.)	8.0 (.)	70.0 (.)	50.0 (.)
*p*-value	0.970	0.936	0.841	0.695	0.260	0.732	0.836	0.906	0.423	0585	0.299	0.971
**Metastatic Disease**												
No	54.3 (17.3)	64.6 (19.3)	−9.7 (11.7)	22.8 (5.2)	37.3 (24.5)	21.1 (4.4)	13.8 (9.9)	25.5 (5.3)	5.6 (3.7)	11.9 (5.6)	57.5 (11.2)	45.2 (25.6)
Yes	58.3 (15.7)	55.4 (11.7)	−10.8 (8.3)	19.2 (4.3)	39.3 (19.0)	19.6 (3.7)	11.0 (8.2)	23.1 (4.9)	8.5 (4.2)	13.9 (5.1)	64.3 (8.5)	42.7 (23.7)
*p*-value	0.517	0.130	0.774	0.055	0.814	0.341	0.428	0.228	0.080	0.599	0.076	0.789
**Cancer Cachexia**												
No	47.6 (16.2)	72.6 (17.3)	0.3 (3.9)	24.8 (5.2)	38.5 (22.8)	21.2 (5.7)	18.1 (10.5)	26.9 (6.7)	3.9 (3.5)	14.4 (6.6)	58.8 (8.3)	38.6 (26.6)
Yes	59.9 (16.7)	56.1 (15.2)	−14.9 (8.1)	19.3 (4.4)	39.4 (22.4)	20.2 (3.6)	10.3 (7.9)	23.4 (4.2)	8.2 (4.0)	12.0 (4.7)	63.3 (10.8)	41.4 (21.8)
*p*-value	0.094	0.023	0.000	0.012	0.930	0.584	0.055	0.130	0.022	0.337	0.300	0.787
**NST**												
Low risk	49.7 (23.3)	69.0 (22.1)	−5.0 (4.3)	23.0 (5.6)	57.1 (31.6)	23.4 (5.5)	13.1 (9.4)	27.5 (6.3)	0.8 (1.1)	14.8 (3.6)	64.0 (11.4)	30.4 (96.6)
Medium risk	44.5 (12.1)	66.0 (15.5)	−6.1 (8.3)	22.3 (5.4)	31.3 (18.1)	19.1 (3.4)	20.5 (12.3)	25.5 (5.1)	4.7 (1.5)	17.3 (8.6)	57.5 (5.0)	41.3 (25.5)
High risk	58.7 (15.3)	59.0 (15.5)	−13.1 (8.0)	20.4 (5.2)	36.8 (16.7)	20.5 (3.0)	11.2 (8.1)	24.0 (3.8)	9.5 (2.4)	10.6 (4.7)	63.7 (10.2)	47.6 (26.4)
*p*-value	0.258	0.462	0.082	0.595	0.134	0.208	0.220	0.354	0.000	0.082	0.517	0.385
**PG-SGA**												
At risk	55.4 (15.6)	61.7 (19.3)	−10.8 (9.6)	21.0 (5.9)	38.5 (24.0)	20.0 (3.0)	13.3 (10.9)	24.2 (1.7)	8.9 (3.6)	7.3 (0.9)	62.9 (9.5)	50.9 (33.6)
Severely malnourished	53.1 (17.6)	61.2 (17.2)	−10.5 (9.7)	21.6 (5.0)	40.9 (22.2)	21.1 (4.2)	12.5 (8.1)	25.0 (5.4)	5.6 (4.4)	14.6 (4.8)	62.0 (9.8)	35.0 (18.6)
*p*-value	0.767	0.952	0.936	0.807	0.819	0.572	0.854	0.715	0.102	0.001	0.985	0.158
**KPS**												
<50	58.7 (14.4)	58.4 (17.2)	−9.3 (14.6)	21.0 (4.9)	24.6 (9.2)	18.9 (3.8)	11.9 (9.9)	22.7 (5.8)	6.4 (3.8)	13.5 (7.7)	47.8 (4.4)	55.5 (26.8)
≥50	55.1 (17.4)	61.1 (16.7)	−10.6 (8.2)	21.1 (5.3)	44.1 (22.6)	20.9 (4.1)	12.5 (8.8)	24.9 (4.8)	7.1 (4.3)	12.1 (4.4)	66.2 (6.6)	39.6 (22.4)
*p*-value	0.586	0.684	0.786	0.988	0.028	0.249	0.877	0.316	0.724	0.587	0.000	0.117
**WHO-PS**												
1	52.5 (27.3)	48.5 (6.9)	−11.2 (5.5)	17.8 (2.5)	35.2 (12.0)	19.3 (2.4)	15.3 (10.8)	24.1 (2.3)	10.7 (4.0)	12.3 (3.8)	73.3 (5.7)	28.0 (7.2)
2	58.7 (16.0)	58.3 (16.7)	−14.5 (10.8)	20.4 (5.0)	42.1 (25.0)	20.7 (4.5)	11.0 (8.6)	24.1 (5.2)	7.5 (4.3)	11.9 (4.4)	64.6 (7.8)	36.3 (22.0)
3	52.9 (14.5)	63.9 (17.6)	−5.0 (8.5)	22.3 (5.6)	35.4 (21.1)	20.1 (4.4)	13.8 (9.6)	24.5 (6.1)	5.2 (3.5)	13.1 (6.7)	55.4 (8.7)	55.9 (25.8)
4	77.8 (.)	75.1 (.)	−7.9 (.)	21.5 (.)	33.9 (.)	21.8 (.)	4.0 (.)	23.1 (.)	8.0 (.)	∙	40.0 (.)	49.0 (.)
*p*-value	0.457	0.394	0.169	0.548	0.889	0.944	0.655	0.995	0.213	0.883	0.001	0.132

*t*-test and ANOVA have been performed accordingly to detect differences in the mean values of measurements in the different groups. Values are presented as the mean (SD) for each nutritional, PS and QoL measure by key characteristics. BMI = body mass index, CRP = C-reactive protein, FACT-G = functional assessment of cancer therapy-general, HGS = hand grip strength, KPS = Karnofsky performance status, MAC = mid arm circumference, MUAMC = mid upper arm muscle circumference, NST = nutritional screening tool, PG-SGA = patient-generated subjective global assessment, PN = parenteral nutrition, TSF = tricep skinfold thickness, WCC = white cell count, and WHO-PS = World Health Organisation performance status. Type of malignancy: gynaecological: breast and endometrial; lower gastrointestinal: small bowel, colon, and sigmoid; upper gastrointestinal: oesophageal and gastric; haematological: leukaemia, lymphoma, and amyloidosis; other: penis and bladder.

**Table 4 nutrients-12-02357-t004:** Cronbach’s alpha coefficient for FACT-G and HPN-QoL factors.

FACT-G Subscales	Alpha	Mean (SD)
Physical well-being	0.605	11.0 (5.8)
Social/Family well-being	0.818	22.8 (5.60
Emotional well-being	0.567	14.7 (6.2)
Functional well-being	0.840	10.8 (6.6)
HPN-QoL factors		Median (IQR25-IQR75)
Pain	0.88	−0.06 (−1.13–0.82)
Worrying	0.80	0.17 (−0.91–0.71)
Walking/socialising	0.81	−0.34 (−0.66–0.60)
Energy/independence	0.84	−0.10 (−0.63–0.73)
Activities	0.81	−0.27 (−0.83–0.75)

Alpha = Cronbach’s alpha coefficient, IQR = interquartile range, FACT-G = functional assessment of cancer therapy-general, and HPN-QoL = home parenteral nutrition quality of life questionnaire.

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
