# Peer review of "Nutritional Status and Quality of Life in Hospitalised Cancer Patients Who Develop Intestinal Failure and Require Parenteral Nutrition: An Observational Study"

_nutrients, 2020, doi:10.3390/nu12082357_

Round 1

Reviewer 1 Report

Overall impression of the study is that is an unfinished manuscript. The title does not correlate with the aims. It may be a difficult challenge to use the design chosen to draw any conclusions from this study. By reading the study it is not clear to me what the authors wants to tell us; to describe recent practice? To make a description of a limited patient population who did receive PN? To describe / compare methods in daily clinical practice, or to forward clinically relevant advice about the use of nutritional screening tools based on a very limited study population with noticeable heterogeneity.

It seems to me that the chosen study format is not appropriate for the purpose as well I find it to be an unclear presentation of data including unexplained missing data, tables without headings / explanations.   

Abstract

Headings are missing – it’s hard to get a quick overview of the study without the headings. Study design is not well described in the abstract.

Introduction

Line 62-63;

It is important to stress that PN should not always be an option in patients with advanced cancer. It may be an option, if patients want this type of nutrition, and are willing to cope with the practicalities that go with it (management, intravenous catheter, risk of serious infections). The effect of PN in patients with advanced cancer remains unclear.

Line 85 Abbreviation; NS; first time use??? Please write it out.

2.2 Study population

It is a large mix of patients with very different needs of nutrition included in the study. It may have be useful to present patient characteristics according to ESPEN, pathophysiological definitions of intestinal failure.https://www.ncbi.nlm.nih.gov/pubmed/?term=ESPEN+endorsed+recommendations.+Definition+and+classification+of+intestinal+failure+in+adults.

2.3 Study collection

The definition of Cachexia used in the study is not sufficient; and not including sarcopenia.

  1. Results

From the text and the table 1 it is hard to read the characteristics for all the patients. There is a lot of missing data, un-explained. Data are presented for variating from 18 to 30 patients. Why?

Indications for parenteral nutrition in this dataset are variable, patients were described with cachexia (60%) and 46 % with a low performance (<2) at the onset of PN. It is difficult to read if the patients were offered PN as best supportive end of life care and if they were discharged. Or if the study population was patients offered nutrition aiming at improvement of the nutritional status.

Table 2 +3

Missing subtitles and descriptions of the data.

Discussion.

Nutritional status; why not include the Muscle mass measures as golden standard; data must be available retrospectively?

4.1.1

Line 363-364 Most important determinator of malnutrition in these patients may be the stage of cancer disease? – or the pathophysiologic condition of intestinal failure?

4.1.2

Anthropometrical indexes; to see an effect of PN on these measures, it seems very important that the patients do not have refractory cachexia with some degree of sarcopenia, beyond the therapeutic window of nutrition. Patients with a bad performance are not expected to have effect of nutritional therapy https://www.ncbi.nlm.nih.gov/pubmed/21296615

Author Response

Please find a word document with the response attached.

Reviewer 2 Report

This manuscript aims to assess nutritional status, function and quality of life in patients with cancer and an indication to start parenteral nutrition.

The overall quality of English is good, the tables contribute to the paper and are formatted well in general.

I had trouble finding a logical train of thought in this manuscript. To me it seemed like values were taken at an internal audit of the clinic and the authors are now trying to evaluate all of them in one long manuscript. I suggest to focus the manuscript on 2-3 specific research questions and eliminate some of the side topics.

The manuscript is overall very wordy and a lot of text is redundant with the tables. I therefore suggest extensive reduction of length and only focusing on the main findings in the text.

The results section requires a major revision, as it is written neither objectively nor consistently in the correct tense and includes a lot of methods, thoughts for the discussion and editorial comments.

In my personal opinion, the extend of the analyses are too great for the small sample size of only 30 patients.

The title of the manuscript is misleading: the dataset gives basically a baseline status of patients with advanced cancer and does not reflect any treatment effect, as the patients are just starting on PN.

Specific comments:

  • I suggest structuring the abstract into background, methods, results and conclusion. Please make clear what type of study this is in abstract & methods.
  • Avoid too many abbreviations in the abstract. I am unclear what the format “[60.3 (16.6) kg] „ means
  • Entire manuscript: please make sure to distinguish between malnutrition, sarcopenia and cachexia, as they are intertwined but separate identities. If you use a muscle parameter such as MUAMC, please explain why it was regarded as a nutrition parameter.
  • Line 55: could you make a statement about the use of ultrasound & BIA for the assessment of muscle mass in these patients?
  • Section 2.3: please make sure to introduce all abbreviations. I am unclear which parameter was obtained at which time during the study, consider adding a table for easier understanding?
  • In line 96 it is stated that these patients were started on PN, Line 168 seems to contradict this statement, could you please clarify?
  • Lines 162-181 are redundant with the tables, I suggest eliminating or drastical shortening of this section
  • Table 2: why give mean (SD) and median (IQR)? This is not a common approach and contradicts the described methods
  • Section 3.2.3 please stick to the correct tense and remain objective in your description of results. The whole section is redundant with table 3, please shorten to avoid excessive length
  • Line 240: correlated with what?
  • Table 3: I do not understand what the authors did here. Please give a more precise table description
  • Section 3.3.1. please refrain from all the editorial comments in the results section, remove lines 251-260. Lines 261-268 can be shortened dramatically, please only include your results and eliminate comments and methods.
  • Section 3.5 text redundant with figure, consider moving the majority of text to methods or eliminate
  • Section 4: In my opinion this needs to be shortened dramatically and adapted to the focus of the manuscript after major revision
  • Line 533: I suggest leaving the self-assessment of the design of study quality out? As I understood, the data were obtained as part of an audit, not as part of a prospective and registered study.
  • Line 536: what were the mean inter- and intrarater variabilities?
  • Section 5: The manuscript does not make clear in which way the clinicians and patients benefit from this analysis

Author Response

Please find a word document with the responses attached.
